# Genome-Wide Identification and Expression Analysis of the Strawberry *FvbZIP* Gene Family and the Role of Key Gene *FabZIP46* in Fruit Resistance to Gray Mold

**DOI:** 10.3390/plants9091199

**Published:** 2020-09-14

**Authors:** Bei Lu, Yuanhua Wang, Geng Zhang, Yingna Feng, Zhiming Yan, Jianhua Wu, Xuehao Chen

**Affiliations:** 1School of Horticulture and Plant Protection, Yangzhou University, Yangzhou 225000, China; yzulubei@163.com; 2Department of Agronomy and Horticulture, Jiangsu Vocational College of Agriculture and Forestry, Jurong 212400, China; wangyuanhua0511@163.com (Y.W.); gengzhang@jsafc.edu.cn (G.Z.); fengyingna88@126.com (Y.F.); yanzhim@jsafc.edu.cn (Z.Y.); 3Engineering and Technical Center for Modern Horticulture, Nanjing 210000, China

**Keywords:** strawberry, bZIP transcription factors, bioinformatics, *FabZIP46*, gray mold disease

## Abstract

A total of 54 *FvbZIP* genes were identified from the strawberry genome. These genes were found to be unevenly distributed on seven different chromosomes, and two of the genes had no matching chromosomal localization. *FvbZIP* genes were divided into 10 subfamilies according to protein sequence, and the structures of these genes were found to be highly conserved. Based on the bioinformatics analysis of *FvbZIP* genes, the expression of *FabZIP* genes changed during different stages of its growth and of its infection with gray mold disease. *FabZIP46* was substantially upregulated, and its expression remained relatively high. *FabZIP46* was cloned from cultivated strawberries by homologous cloning. The results of a transient transgenic assay revealed that the damage to the fruit tissue was markedly alleviated in strawberries overexpressing *FabZIP46*, with the incidence rate being substantially lower than that in the control group. By contrast, a brief silencing of *FabZIP46* had the opposite effect. The results revealed that *FabZIP46* played a positive role in the resistance of strawberries to *Botrytis cinerea*. The study findings provide valuable insights into the role of bZIP transcription factors as well as a theoretical reference for the regulation of resistance to gray mold disease in strawberry fruit.

## 1. Introduction

The strawberry (*Fragaria* × *ananassa*) is the most cultivated among the small berries in the world and has both ornamental and nutritional value. Strawberries are currently one of the main crops under protected cultivation, and China is one of the largest strawberry producers in the world. Owing to its small size, short fruit growth cycle, and high genetic transformation efficiency, the strawberry is a suitable model plant for fruit research [1,2]. Therefore, it plays a pivotal role in both commercial production and scientific research in the vegetable and fruit industry. However, the common cultivated strawberry is octoploid, so its genetic background is very complex, but via the diploid woodland strawberry (*Fragaria vesca*) we have completed the work of genome sequencing. The published strawberry genome sequence is valuable for identifying ideal agronomic traits and stress resistance genes in the strawberry at the genome level. The strawberry production industry in Jiangsu Province focuses on forcing cultivation, wherein the growth period lasts from September to April and the fruiting period lasts from December to April. The low temperature and low photoperiod during the fruiting stage have huge impacts on the growth and development of strawberries in that the plants are more susceptible to gray mold disease, which can severely affect the yield and quality of strawberries. Gray mold disease is a critical concern in agricultural production worldwide and caused by *Botrytis cinerea* Pers. *B. cinerea* is considered a critical pathogenic and necrotrophic fungus. A genomic analysis revealed that *B. cinerea* secretes approximately 40 types of toxins and can infect more than 200 different plant species both before and after harvesting, resulting in severe economic losses to agricultural production [3,4,5]. Gray mold disease is one of the most serious diseases in strawberries and affects as much as 60% of the fruit once infected. The infection may occur both before and after fruit harvesting, thus causing the fruit to rot and become inedible. In severe cases, infection can lead to a total loss of yield. Thus, gray mold disease greatly impedes the development of strawberry production [6].

Studies have indicated that transcription factors (TFs) are key regulators of gene expression and play a pivotal role in the complex molecular defense network of plant immunity [7,8]. Basic leucine zippers (bZIP) are a group of transcription factors commonly found in animals, plants, and microorganisms. They have a basic amino acid region composed of 60–80 amino acid residues and a highly conserved bZIP domain composed of leucine zippers [9]. Numerous studies have indicated that members of the *bZIP* gene family are involved in biological processes such as growth and development, senescence, hormone regulation, energy metabolism, pathogen defense, and abiotic stress in plants [10,11,12,13,14]. For example, *ZmbZIP4* overexpression in maize can markedly promote the growth and development of roots. Chromatin immunoprecipitation revealed that *ZmbZIP4* directly binds to the promoters of genes that are involved in root developments, such as *ZmLRP1* and *ZMSCR*, thereby promoting their transcription [15]. Transcription factor *CabZIP1* is expressed only in the roots and flowers of hot peppers, but not in the leaves, stems, and fruits. *Arabidopsis* transfected with *CabZIP1* exhibits a dwarf phenotype. Therefore, this gene is inferred to be involved in plant growth and development [16]. Overexpression of *CabZIP1* in *Arabidopsis* can enhance the resistance of the plants to *Pseudomonas syringae* pv. tomato (DC3000) and can induce the expression of resistance-related genes *AtRP4* and *AtRD29A* in *Arabidopsis* [13]. *OsbZIP71* in rice can directly bind to the promoters of abiotic stress-related genes *OsNHX1* and *COR413-TM1* to activate their transcription, thereby enhancing the resistance of the plant to drought and salt. This finding supports the importance of *OsbZIP71* in abiotic stress responses [17]. Zhang et al. isolated *LrbZIP1* from lily plants and found that overexpression of *LrbZIP1* in tobacco markedly inhibited the growth of *Fusarium oxysporum* [18]. All of the above studies suggest that *bZIP* genes can function alone, collaboratively, or in a complex to regulate plant growth and development or to induce a defense response against pathogens.

Although the role of many members of the *bZIP* gene family has been widely investigated in various crops, little is known about their defense-related functions and regulatory mechanisms in strawberries. Among the identified *FvbZIP* transcription factors, Wang et al. defined those associated with drought and heat stress responses in strawberries [19]. No specific genes have been identified as playing a role in the disease resistance mechanisms of the strawberry. Therefore, to further investigate the role of the *bZIP* gene family in the disease resistance of strawberries, we conducted a bioinformatic analysis to analyze the strawberry genome from more perspectives; to systematically and comprehensively analyze the basic physical and chemical properties of strawberry *FvbZIP* genes; and to understand the subclassification, evolutionary relationships, chromosomal localization, and conservation of domains of the encoded proteins. Based on the bioinformatics analysis of *FvbZIP* genes in the woodland strawberry, we also explored the changes in the expression of octoploid strawberry *FabZIP* genes at seven stages after infection with *B. cinerea.* Based on the study of *bZIP* genes in the woodland strawberry, we expected to screen potential candidate genes related to disease resistance and verify the role of related genes in gray mold disease in cultivated strawberries. The study findings may provide theoretical support for disease prevention and development of disease resistance for strawberry breeding in the future.

## 2. Results and Analysis

### 2.1. Members of the bZIP Transcription Factor Family in Strawberries

By using the conserved domain of bZIP as a probe, aligning the annotated genome, and manually deleting repetitive and redundant sequences, 54 transcription factors of the *bZIP* gene family were identified in the woodland strawberry. Four more genes were added compared to previous studies by Wang et al. [19]. According to the chromosomes on which these genes are located, all the *FvbZIP* transcription factors were systematically numbered from *FvbZIP1* to *FvbZIP54* (Appendix A). Related information, including genetic mapping, initiation site, termination site, and protein sequence length, was also categorized. The number of amino acids in FvbZIP proteins ranges from 114 to 847, with an average of 370. FvbZIP31 is the smallest bZIP protein and has 114 amino acids (Appendix A). By using the online tool ProtParam, the molecular mass of the 54 FvbZIP proteins was predicted to range from 13.51 to 93 kDa, and the isoelectric point ranged from 4.48 to 10.61 (Appendix A). Moreover, protein subcellular localization prediction revealed that *FvbZIP* genes were principally located within the nucleus.

### 2.2. Phylogenetic Relationships among the Family Members of bZIP Transcription Factors in Strawberries

A comparison of multiple genome sequences allows researchers to investigate species that are understudied by providing information from the research of species that have been extensively studied [20]. To study the evolutionary relationship between *FvbZIP* genes in the strawberry and those in *Arabidopsis* and rice, a phylogenetic tree was constructed based on the amino acid sequences of strawberry, *Arabidopsis*, and rice bZIP proteins was constructed by using the software MEGA7.0 (Figure 1), and a separate phylogenetic tree of strawberry *FvbZIP* genes was constructed for reference (Figure 4). By referring to the clustering method of Jakoby et al., which classified the *bZIP* gene family according to the domain characteristics, the *bZIP* genes of strawberry, *Arabidopsis*, and rice were classified into 10 subfamilies, namely A, B, C, D, E, F, G, H, I, and S, by using a phylogenetic tree [21]. The analysis showed that 85% of the 54 FvbZIP proteins cluster predominantly with *Arabidopsis* bZIP proteins, which indicates their closer evolutionary relationship than with rice bZIP proteins. Both strawberries and *Arabidopsis* are dicotyledons and are thus more closely related and may share an evolutionary ancestor. According to the kinship division in the phylogenetic tree, the function of strawberry *FvbZIP* genes can be preliminarily predicted by referring to their homologous genes in *Arabidopsis*. *FvbZIP* genes were also divided into 10 subfamilies in the phylogenetic tree of the *bZIP* gene family of the strawberry, and the genes of the same subfamilies were clustered more closely together. Among these subfamilies, the proportion of members in subfamilies A, D, and S had a similar number of members; subfamily A was the largest with 12 members, and subfamily B was the smallest with only one member. In similar studies, the *bZIP* gene families of the grape and apple have also been classified into 10 categories [22,23]. No species-specific populations were found in the phylogenetic tree of strawberry *FvbZIP* genes. Despite the differences within each subfamily of different species, members within the same subfamily are fundamentally similar, which suggests that the evolution of *bZIP* genes in plants is relatively conserved.

### 2.3. Chromosomal Localization and Analysis of the Gene Structure of Strawberry bZIP Genes

According to the annotation information provided by NCBI, the chromosomal localization of the 54 genes was visualized through MapGene2Chromosome2. The results revealed that 52 genes were unevenly distributed on seven chromosomes and two genes, *FvbZIP53* and *FvbZIP54*, had no matching chromosomal localization (Figure 2). Chromosome 2 contained the largest number of genes, 17. During a BLAST search of the structural domain of *FvbZIP* genes in the NCBI database, nine FvbZIP proteins were found to not only contain the typical bZIP conserved domains, but also other domains. Among these, FvbZIP35, FvbZIP37, and FvbZIP50 had the MFMR domain at the N-terminus of the bZIP domain. The MFMR domain has been reported to play a critical role in regulating protein–protein interactions [24]. FvbZIP16, FvbZIP26, FvbZIP39, FvbZIP41, FvbZIP47, FvbZIP51, and FvbZIP53 had a DELAY OF GERMINATION 1 (DOG1; PF14144) domain in addition to the bZIP conserved domain. DOG1 of *Arabidopsis* has been reported to be involved in the regulation of seed dormancy [25].

Multiple sequence alignment visualization was performed on the amino acid site in the amino acid conserved domain of the bZIP proteins of the strawberry. The results revealed that the N-terminus of the bZIP conserved domain of FvbZIP proteins exhibited a N-X7-R/K motif, which formed the basic region of bZIP. In addition, one leucine was present for every seven other amino acids in the C-terminal direction after the nine amino acids of R/K, which formed the leucine zipper domain (Figure 3). The basic domain of FvbZIP32 was composed of Lys instead of Asn, which was consistent with that in the previously reported rice proteins *OsbZIP21* and *OsbZIP82* [26]. From Appendix A, it is evident that the bZIP conserved domains of *Arabidopsis* and strawberry proteins are almost identical, suggesting that, genetically, the strawberry is more closely related to *Arabidopsis* than to rice. These results indicate that the role of strawberry *FvbZIP* genes might be similar to that of *Arabidopsis bZIP* genes.

To further investigate the relationship among the protein components of the *bZIP* family, the online prediction tool MEME was used to predict 20 conserved motifs of the strawberry *bZIP* family (Figure 4). The results showed that all the proteins in the strawberry *bZIP* family contained 1–7 motifs. FvbZIP10 exhibited only one motif; additionally, all the proteins, except FvbZIP32, exhibited Motif 1. Although the proteins encoded by FvbZIP proteins contained 114–847 amino acids, the MEME diagram of the protein motif structure indicated that the clustered FvbZIP proteins had the same protein motif location and almost the same types of conserved elements. Although the functions of most of the motif elements were unknown, certain motifs were observed in specific bZIP members, suggesting that these transcription factors had specific functions. A similar finding has been reported in other species, indicating that some elements are relatively conserved evolutionarily [27]. These unknown elements may have specific functions in each subfamily, which may provide a new reference for the functional divergence among members of each subfamily.

Different combinations of exons and introns can exhibit different gene functions. The gene structure presented in Figure 4 can explain the evolutionary relationship of gene families [28]. The number of introns in the 54 *FvbZIP* genes of the strawberry ranged from 0 to 11. Studies have shown that 19% of the genes in the bZIP families of sorghum and rice had no intron, and only two genes in the grape *bZIP* gene family had no intron. However, 11 *FvbZIP* genes were found to have no introns, accounting for 20% of all *FvbZIP* genes, which is higher than that observed in grapes, sorghum, and rice [22,26,29]. Moreover, *FvbZIP* genes with a similar structure were predominantly clustered together.

### 2.4. Response of the FabZIP Genes of Fragaria × Ananassa “Benihoppe” Strawberry to Stress Caused by B. Cinerea

Based on the bioinformatics research of *FvbZIP* genes in the woodland strawberry, in order to study the expression pattern of *FabZIP* genes in the cultivated strawberry, fluorescence quantitative detection was carried out for different development stages of strawberries and different onset stages of gray mold. Real-time fluorescence quantitative analysis was performed to collect the expression data of the 54 *bZIP* genes of cultivated strawberries at different developmental stages and different stages of the gray mold disease (Figure 5). The collected expression data of strawberry *bZIP* genes were subjected to cluster analysis, and changes in the gene expression level were compared using the cluster diagram. Most of the 54 *bZIP* genes appeared to be downregulated during the developmental stages, and the trend was flat. Among them, *FabZIP18*, *FabZIP20*, *FabZIP21, FabZIP22*, *FabZIP38*, *FabZIP42*, and *FabZIP53* were highly downregulated, and the expression level was also relatively low at the onset of gray mold disease. However, the expression of most genes was substantially upregulated after *B. cinerea* infection; only a few genes exhibited a flat expression trend before and after infection. Currently, studies have verified that some members of the S and D subfamilies of bZIP transcription factors are involved in the regulation of plant growth and disease defense mechanism [30,31]. Members of the S subfamily exhibited an inconsistent trend of change in expression levels. With the growth and development of the fruit, only *FabZIP46* was found to be evidently upregulated, whereas the expression levels of other members remained relatively low. Members of the D subfamily, *FabZIP39*, *FabZIP26*, and *FabZIP16*, exhibited a relatively low expression level and a gradual upregulation trend. After infection with *B. cinerea*, *FabZIP27*, *FabZIP36*, and *FabZIP46* of the S subfamily exhibited an apparent upregulation trend at disease onset. Among these genes, *FabZIP46* maintained a relatively high expression level. The expression levels of *FabZIP16*, *FabZIP26*, *FabZIP39*, *FabZIP47*, *FabZIP41*, and *FabZIP51* but not *FabZIP 53* of the D subfamily were evidently upregulated.

From what has been discussed above, *FabZIP46* was substantially upregulated, and its expression remained relatively high at different developmental stages as well as at different stages after the onset of gray mold disease. Therefore, *FabZIP46* was inferred to play a role in the resistance mechanisms of the strawberry to gray mold disease.

The bZIP transcription factors used in the comparison were selected according to a previous report and included AtbZIP2, AtbZIP10, and AtTGA5 in *Arabidopsis* [18], CabZIP1, CabZIP2, and CcPPI1 in hot peppers [13,16,32], OsbZIP1 in rice [33], NtbZIP60 and NtTGA2.2 in tobacco [34,35], VvbZIP23 in grapes, TabZIP1 in wheat, LrbZIp1 in the lily, and SLAREB1 in tomatoes [18,36,37,38]. The results revealed that FabZIP46 had a high homology with LrbZIP1 and CcPPI1. The overexpression of *LrbZIP1* in tobacco can significantly inhibit the growth of *Fusarium oxysporum*. CcPPI1 is a nuclear protein capable of activating transcription in yeast [18]. It is a transcription factor that can be part of the plant defense response against pathogen infection [32]. Therefore, the function of *FabZIP46* may be similar to that of *LrbZIP1* and *CcPPI1* and may be related to the resistance of strawberries to gray mold disease.

### 2.5. Analysis of the Transient Expression of FabZIP46 in Strawberries

*FvbZIP* and *AtbZIP* genes are not located in independent branches. In *“Benihoppe”*, *FabZIP46* has high homology with *bZIP* genes related to disease resistance in other species, indicating that, although gene sequences of different species are different, the bZIP protein has similar functions in regulating signal transduction of plant biological and nonbiological stress response. Therefore, it can be concluded that the *FabZIP46* gene also plays an important role in resistance to pathogenic bacteria in cultivated strawberries. *FabZIP46* was cloned from “*Benihoppe*” by homologous cloning. To further analyze the role of *FabZIP46* in the incidence of gray mold disease in strawberries, *FabZIP46* in strawberries was overexpressed and silenced using the *Agrobacterium*-mediated transient transformation method.

The *FabZIP46* gene in strawberries was transiently overexpressed and silenced after the transient transgenic transformation (Figure 6). The results revealed that the timing of disease onset in strawberries overexpressing *FabZIP46* was markedly delayed compared with that in the empty vector control. The disease incidence rate was also reduced, and the number of diseased fruits remained low even on the sixth day of *B. cinerea* inoculation. By contrast, the timing of disease onset was advanced, and the condition of the fruits was more severe after *FabZIP46* silencing. The fruits injected with different vectors were compared with those inoculated with *B. cinerea* during the same period, and the statistical data revealed that the incidence rate of strawberries overexpressing *FabZIP46* was substantially lower than that of the control group. Most fruits had only small lesions, and almost no fruit had severe lesions on the sixth day after injection. By contrast, the disease incidence rate of the fruits was higher than that of the control group after *FabZIP46* was silenced. The results revealed that nearly all fruits were diseased on the sixth day after injection, and severely affected fruits accounted for more than half of the total fruit (Appendix A). The results suggested that overexpression of *FabZIP46* increases the resistance of strawberries; it not only delayed the onset timing of gray mold disease in strawberries but also reduced the incidence rate. Moreover, the disease onset condition in fruit with silenced *FabZIP46* was not apparently different from that in the control group. The experimental results indicated that *FabZIP46* played a role in the resistance of strawberries to gray mold disease.

### 2.6. Effect of Overexpression and Silencing of FabZIP46 on the Expression of Disease-Resistant Genes

In order to explore the molecular mechanism of *FabZIP46* in resistance to gray mold in strawberries, some genes related to resistance reported in the literature were selected for expression detection. Transcription factors are closely related to plant disease resistance. In this experiment, *FaWRKY1*, *FaWRKY33*, and *FaWRKY70* related genes in the WRKY transcription factor family identified from the disease resistance correlation in plants were selected [39,40,41]. *FaWRKY1*, *FaWRKY33*, and *FaWRKY70* were evidently regulated when *FabZIP46* was overexpressed or silenced. When *FabZIP46* was overexpressed, the expression of *FaWRKY1* and *FaWRKY70* was positively regulated and that of *FaWRKY33* was negatively regulated (Figure 7).

In addition, polygalacturonase-inhibiting proteins (PGIP) and chitinase (CHI) play defensive roles in the disease resistance of strawberries in the plant immune system [42,43]. The pathogenesis-related proteins (PR) also play an important role in the pathogenesis and necrosis of plants [44]. Therefore, *FaPR1*, *FaPR4*, *FaCHI2*, *FaCHI3*, *FaCHI4*, *FaPGIP1*, and *FaPGIP2* were selected for gene expression detection. The results showed that the expression levels of most of these defense genes were also regulated to varying degrees, of which *FaPR1* was strongly regulated.

## 3. Discussion

Numerous studies have indicated that members of the *bZIP* gene family play a role in plant growth and development and respond to biotic and abiotic stresses [45]. Presently, genes related to the *bZIP* gene family members have been extensively identified in *Arabidopsis*, rice, and other model plants [46,47]. Therefore, the role of *bZIP* genes in strawberries was investigated through a bioinformatic analysis. A total of 54 *bZIP* genes of the strawberry were identified in this study, which was less than the number of *bZIP* genes in *Arabidopsis thaliana*. By using the results of cluster analysis on *bZIP* gene members of *Arabidopsis* and rice, the *bZIP* gene members of strawberries were classified into 10 subfamilies. To date, several studies on bZIP transcription factors in plants have demonstrated that different subfamilies control various transcriptional regulation pathways in different plants. Among them, *bZIP* genes of subfamily A are involved in the ABA and stress regulatory network of seeds and plant tissues [11,48]; *bZIP* genes of subfamily C regulate protein storage in the seeds, environment and disease defense, and stress response [49,50]; and *bZIP* genes of subfamily D participate in the plant disease defense and physiological growth process [48,51,52]. Studies on the function of *bZIP* genes of subfamily E are relatively rare, and no functional data are available for members of group E [21]. *bZIP* genes of subfamily G regulate light-regulated signal transduction and seed maturation [53,54]. *bZIP* genes of subfamily H play an indispensable role in light morphogenesis and light signal transduction and are required for the induction of key enzymes and nitrates in nitrogen assimilation [55]. *bZIP* genes of subfamily I play a critical role in the vascular development of plants [56]. Finally, as a comparatively large group, *bZIP* genes of subfamily S play a vital role in disease defense and stress response [57]. Studies on some subfamilies, such as B and F, have not been reported, so further research is required to explore their functions.

Wang et al. analyzed the evolutionary patterns of the homologous *bZIP* genes in strawberries, apples, and peaches [58]. By analyzing the evolution of *bZIP* genes in three species of *Rosaceae*, it is shown that many *bZIP* genes are produced by gene replication. The *FvbZIP* genes were found to have undergone a dramatic evolutionary diversification, with protein sequence similarities ranging from 9% to 100%. In the present study, bZIP proteins from three different plant species were divided into 10 groups, with each group containing at least one *AtbZIP* or *OsbZIP*, indicating the conservation of *bZIP* genes in different plants. This suggests that *FvbZIP* genes share some common ancestor genes with those of other plants. The phylogenetic tree constructed with the proteins of strawberries, *Arabidopsis*, and rice showed that most of the *FvbZIP* genes were more closely related to *Arabidopsis*. This suggests that the evolution of the strawberry *bZIP* gene family is comparatively conserved and these genes may have functions similar to those of *Arabidopsis* homologous genes. Through chromosomal localization, 54 *FvbZIP* genes were found to be unevenly distributed on seven chromosomes, among which *FvbZIP53* and *FvbZIP54* could not be located on the chromosome. The analysis of gene structure revealed that members of the same strawberry *bZIP* subfamily were found to have the same or similar genetic structures, particularly the number and length of exons, indicating that the classification in this study is accurate. The above results indicate that genes in the strawberry *bZIP* gene family are highly conserved in structure, and their function and characteristics are similar to those of *Arabidopsis bZIP* genes. This finding may serve as a reference for the future application of genetic engineering and genetic analysis to verify the function of *bZIP* gene family in strawberries.

The diploid woodland strawberry is not widely cultivated in the strawberry production industry. At present, the main edible strawberry is the octoploid strawberry, so this experiment used “*Benihoppe*” as the research object. The results of real-time fluorescence quantitative analysis on the expression of the 54 strawberry bZIP transcription factors at different developmental stages of the fruits and different stages of gray mold disease showed that *FabZIP46* was substantially upregulated both with the fruit growth and development and with the onset of gray mold disease, and the expression was maintained at a relatively high level. In addition, a phylogenetic tree was constructed by comparing the protein sequences of *FabZIP46* and those of bZIP transcription factors to investigate disease resistance in various species. According to previous reports, *bZIP* genes related to the disease resistance of different species such as tobacco, *Arabidopsis*, rice, peppers, and tomatoes were selected for homology comparison with *FabZIP46* (Appendix A). The results revealed that *FabZIP46* had a high homology with *LrbZIP1* and *CcPPI1.* Therefore, the function of *FabZIP46* may be similar to that of *LrbZIP1* and *CcPPI1* and may be related to the resistance of strawberries to gray mold disease.

bZIP transcription factors are known to be related to abiotic and biotic stress responses. Presently, numerous bZIP transcription factors with specific functions in plants have been identified. Studies have shown that overexpressing or silencing *bZIP* genes may enhance the resistance of plants to abiotic and biotic stresses. Silencing the *CabZIP2* gene in hot peppers results in disease-susceptible traits, and overexpression *CabZIP2* in *Arabidopsis* enhances their resistance to *Pseudomonas syringae* pv. Tomato [16]. Overexpressing *NtTGA2.2* in tobacco may enhance the expression of early genes *PR1-a* and *ParA*, but the expression of early genes could not be induced in SA-treated mutant plants, indicating that *NtTGA2.2* plays a positive regulatory role in pathogen defense [59]. The results of the above studies indicate that *bZIP* genes in cultivated strawberries may also have a similar function of defense against pathogenic bacteria. The disease onset in strawberries overexpressing *FabZIP46* was evidently delayed compared with the control group; the incidence rate was also decreased. By contrast, the disease incidence was severe in fruit after *FabZIP46* silencing, and the incidence rate exceeded 50% on the fourth day. Most functional verification studies of *bZIP* genes have been conducted in the model plant *Arabidopsis*. However, the plant characteristics of *Arabidopsis* are considerably different from those of cultivated strawberries. To date, no study has investigated the role of *FabZIP46* in strawberry gray mold disease. Therefore, further investigations are required to reveal the specific role of *bZIP* genes in strawberries.

A study of transcription factors involved in disease resistance revealed that the expression level of three genes in the WRKY transcription factor family, *FaWRKY1*, *FaWRKY33*, and *FaWRKY70*, were significantly changed when *FabZIP46* was overexpressed or silenced. Therefore, it can be speculated that *bZIP* can interact with WRKY transcription factor and play an important role in the plant response to biotic and abiotic stress and various signaling pathways.

Polygalacturonase-inhibiting proteins (PGIP) and chitinase (CHI) play a defensive role in strawberries [42,43]; pathogenesis-related proteins (PR) also play a critical role in the pathogenesis of plant necrosis [44]. In this study, *FaPR1*, *FaPR4*, *FaCHI2*, *FaCHI3*, *FaCHI4*, *FaPGIP1*, and *FaPGIP2* were selected for gene expression detection. The results revealed that the regulation of the expression of these defense-related genes was varied, and *FaPR1* was highly regulated. This finding indicates that the plant defense response may have multiple underlying defense mechanisms. *FabZIP46* is likely to strengthen the defense of strawberries against *B. cinerea* and delay the onset of gray mold disease by upregulating PR and PGIP genes.

In summary, 54 strawberry *bZIP* gene family members were predicted by adopting the bioinformatics approach, and a cluster analysis was conducted using the *Arabidopsis bZIP* gene family. Bioinformatics data, including domain, gene structure, chromosome distribution, and the distribution of conserved elements, indicated that these genes are relatively conserved during evolution and may play a key role in the stress response to gray mold disease. The study results provide insights into the function of the strawberry FvbZIP protein family and may be used as a reference for screening genes related to disease resistance through genetic engineering. The cluster analysis of data from real-time fluorescence quantitative experiment was used to predict that *FabZIP46* is related to the resistance mechanism of strawberries to gray mold disease, and its function was verified using a transient transformation experiment. The results revealed that *FabZIP46* positively regulated the resistance of strawberries to gray mold disease.

## 4. Materials and Methods

### 4.1. Fungi and Plant Materials

The *B. cinerea* used in this study was isolated from infected plants in a strawberry planting base of Jiangsu Agricultural Expo Garden by the Jiangsu Horticultural Modern Technology Engineering Center. After conventional purification treatments, the isolated *B. cinerea* was inoculated on potato dextrose agar (PDA) and cultured at 25 °C for 14 days. After four weeks of culture, the mycelium was scraped from the surface of the strain, and the spores of *B. cinerea* were diluted with sterile water into a spore suspension (containing 0.01% (*v*/*v*) Tween-20) at a concentration of 1 × 10^5^ spores/mL for fruit inoculation.

The test plant, the octoploid cultivar “*Benihoppe*” strawberry, was picked from the strawberry planting base of Jiangsu Agricultural Expo Garden between December 2019 and February 2020. The fruits were uniform in size and shape, with no deformities.

Based on a previous study, the development of strawberries was divided into seven stages according to the local environmental conditions for cultivation [60]: small green fruit (SG, 20–25 d postflowering), medium green fruit (MG, 28–33 d postflowering), large green fruit (BG, 35–40 d postflowering), white fruit (Wh, 40–45 d postflowering), color-turning fruit (Tu, 45–50 d postflowering), half-red fruit (HF, 48–53 d postflowering), and red fruit (Re, 50–55 d postflowering). Ten strawberries at each developmental stage were collected from the field. After the seeds were removed from the surface, the fruits were rapidly frozen with liquid nitrogen and stored in a refrigerator at −80 °C. The expression levels of *FvbZIP* genes were detected in these strawberry samples at different developmental stages. The experiment was repeated three times.

### 4.2. Extraction of FvbZIP Genes from Strawberries and Prediction of Physicochemical Properties

By using the conserved structural domain PF000170 of bZIP as a probe, the whole genome sequence and CDS sequence of *bZIP* genes in the strawberry were collected from the database of the National Center for Bioinformatics (https://www.ncbi.nlm.nih.gov). The structures of the downloaded FvbZIP proteins were predicted using NCBI-CDD. Proteins that did not include the bZIP structural domain were deleted. In addition, the molecular weight and isoelectric point of the amino acid sequence of all bZIP proteins were predicted using the online tool ProtParam (http://web.expasy.org/protparam/), and the subcellular localization of all bZIP proteins were predicted using the online tool Cell-PLoc-2.0 (http://www.csbio.sjtu.edu.cn/bioinf/Cell-PLoc-2/).

### 4.3. Phylogenetic Analysis and Sequence Alignment of Strawberry FvbZIP Genes

The amino acid sequences of *Arabidopsis* bZIP proteins were downloaded from the *Arabidopsis* Information Resource (TAIR, https://www.arabidopsis.org/), and those of rice were downloaded from RiceData (https://www.ricedata.cn/gene). Multiple sequence alignment was conducted to compare the protein sequence of the downloaded protein and that of the predicted FvbZIP proteins by using ClustalX2.0 (http://www.clustal.org/clustal2/). All the parameters were set to the default. The result of the sequence alignment was imported to MEGA7.0 (http://www.megasoftware.net/history.php) to plot a phylogenetic tree by using the neighbor-joining method. Additionally, an independent phylogenetic tree was constructed with the complete FvbZIP protein sequences. The execution parameters were set as follows: mode: “Poisson correction” gap setting: “Complete Deletion” validation parameter: Bootstrap = 1000, and random seeds. The gene mapping information of *FvbZIP* genes was obtained from the NCBI database, and a chromosome mapping graph was plotted using the online software MapGene2Chromosome2 (http://mg2c.iask.in/mg2c_v2.0/).

### 4.4. Analysis of the Structure of FvbZIP Genes in the Strawberry and Analysis of the MEME Protein Motif

Through the website Gene Structure Display Server (http://gsds.gao-lab.org/), the whole genome sequence and CDS sequence of *FvbZIP* genes were formulated into an information map containing the structure of each *FvbZIP* gene. Moreover, the structures of the conserved domain and other protein motifs of strawberry FvbZIP proteins were analyzed. The protein sequence of all the strawberry bZIP proteins were entered into the website, with the highest motif number set to 20 and the other parameters set to the default (http://meme-suite.org/tools/meme).

### 4.5. Real-Time Fluorescence Quantitative Analysis of Gene Expression

All the reagents required for real-time fluorescence quantitation were provided by, and primer synthesis (Appendix A) was conducted by, TAKARA Bio., Inc. (http://www.takara.com.cn/). A total RNA extraction kit was used to extract total RNA from strawberries, and M-MLV reverse transcriptase was used to synthesize cDNA. Quantitative RT-PCR was performed using SYBR PreMix, and the internal reference gene was *FaACTIN*. The relative expression level of the genes was calculated using the 2^−ΔΔCT^ method. The experiment was biologically duplicated three times.

### 4.6. Gene Cloning and Vector Construction

Referring to the information of *FvbZIP46* published by NCBI (GenBank accession number: XM_011468884.1), the full-length *FabZIP46* gene was obtained from “*Benihoppe*” strawberries by homologous cloning. Full-length *FabZIP46* contained 691 bp, encoding 230 amino acids. The vector was constructed using Gateway technology. After verification of the sequence, the cloned fragment was inserted into the TOPO vector (TOPO PCR Cloning technology; Thermo Fisher Scientific Inc., Shanghai, China). Subsequently, an LR reaction between the TOPO vector and the target expression vector of the plant pH7WG2D was conducted to generate *35S::FabZIP46* to obtain a *FabZIP46* overexpression vector (overexpression vector *FabZIP46*, *FabZIP46-OE*). The LR reaction is a recombination reaction between an attL entry clone and an attR destination vector. The LR reaction is used to transfer the destination sequence to one or more destination vectors in parallel reactions. The empty vector pH7WG2D (EV-OE) was used as control. The plant expression vector pH7WG2D was constructed to contain a green fluorescent protein-encoding (*e-GFP*) gene to facilitate further detection. Under a stereoscopic fluorescence microscope, the green fluorescence emitted by successfully transformed strawberries can be directly observed, whereas no green fluorescence can be detected in unsuccessfully transformed strawberries.

For the transient silencing of *FabZIP46* in strawberries, a hairpin structure of *FabZIP46*, *pFGC5941:FabZIP46:RNAi* was constructed using pFGC5941. pFGC5941 is a widely used vector for RNA interference of plant genes. Next, *FabZIP46*:*RNAi* was amplified using a specifier, and the product was subjected to the LR reaction along with the target vector pH7WG2D to generate a *FabZIP46::RNAi* silencing/expression vector (RNAi-expression vector *FabZIP46*, *FabZIP46-RNAi*). The empty vector pH7WG2D (EV-RNAi) was used as a control.

All vectors were transformed into *Agrobacterium GV3101* by using a freeze–thaw method for subsequent transient transformation of strawberries. All the amplified sequences and specifiers used for vector construction are given in Appendix A.

### 4.7. Transient Transformation of Strawberries and Experimental Design

Non-diseased strawberries that were at the stage of turning from white to red and growing on healthy plants were picked from the field. After sterilization, the fruits were placed in a confined space at a temperature of 25 °C and a relative humidity of 85%. Using a 1-mL syringe, a single dose of 1 mL of the prepared bacterial injection solution was injected into the hollow pith of the strawberry through the fruit stalk. The injection volume of the *Agrobacterium* suspension was moderately adjusted according to the size of the fruit to ensure that the bacterial solution completely infected the fruit. A total of 200 fruits were injected with each bacterial carrier suspension, and changes in the fruit were observed and recorded daily after the injection. The vectors used in the transient transformation study carried the green fluorescent protein gene (*eGFP*) with extremely strong expression. Therefore, once successfully transformed, fluorescence was observed in the fruit by using the fluorescence observation system on the second day after injection of bacterial suspension. The fluorescence intensity reached the maximum on the fifth day. On the fourth day after injection and after the fluorescence became observable, three samples were selected from each group to examine the expression level of *FabZIP46*. Subsequently, 150 fruits that exhibited fluorescence were selected from each group for *B. cinerea* inoculation. Holes were pricked on the surface of the selected fruits and *B. cinerea* was inoculated onto the pricked sites to observe the disease incidence. After inoculation with *B. cinerea*, three samples were taken every 24 h to detect the expression level of related genes; the samples were taken six consecutive times. The remaining 132 strawberries were used for statistical analysis of disease incidence. The experiment was repeated three times in the fruiting stage of strawberries. The majority of the strawberries were photographed to record the disease progression during the experiment.

## Figures and Tables

**Figure 1 plants-09-01199-f001:**
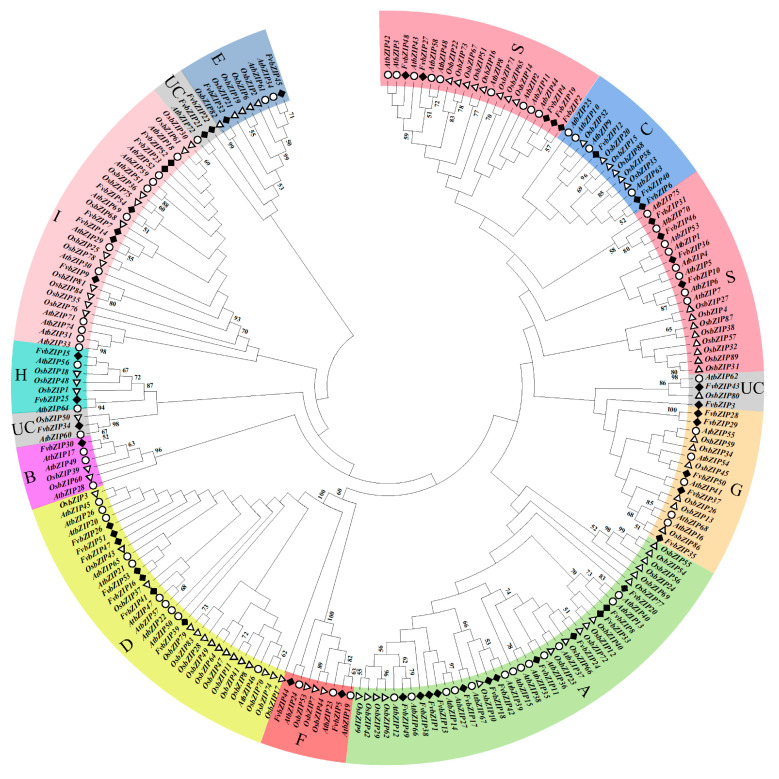
Phylogenetic analysis of bZIP genes. At represented *A. thaliana*, Os represented *O. sativa*. The phylogenetic tree was generated using the amino acid sequences of selected bZIP genes via NJ methods. All strawberry bZIP genes, together with their *A. thaliana* and *O. sativa* homologues, were classified into 10 groups.

**Figure 2 plants-09-01199-f002:**
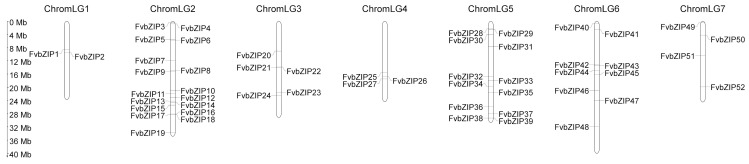
Chromosome distribution of *FvbZIP* genes. The chromosome numbers are demonstrated at the top of each chromosome and the scale is in megabases (Mb).

**Figure 3 plants-09-01199-f003:**
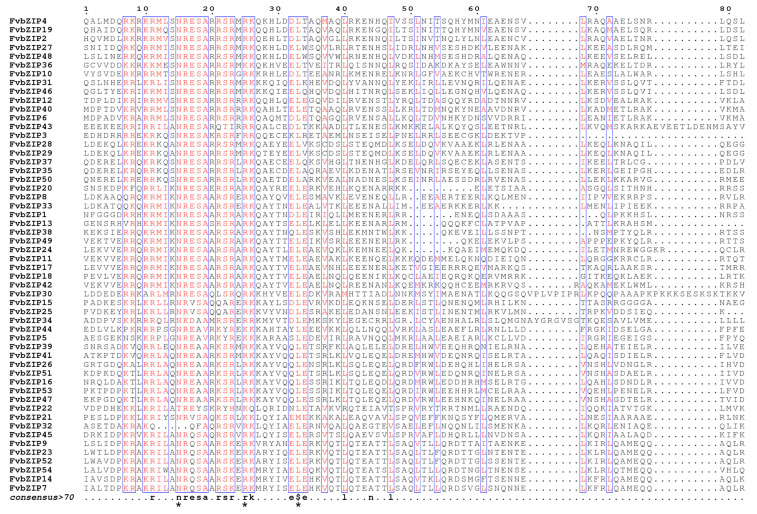
Multiple alignments of conserved domains of all identified FvbZIP proteins. Asterisks show the conserved amino acids of the bZIP domain.

**Figure 4 plants-09-01199-f004:**
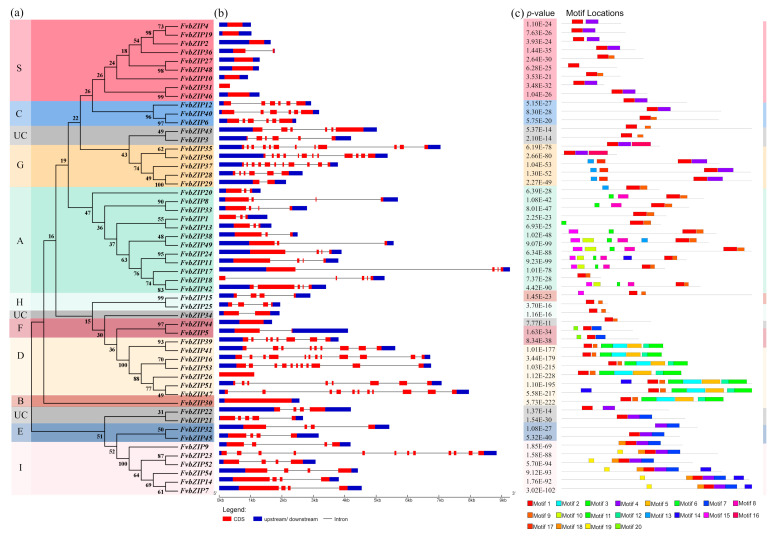
(**a**) Phylogenetic analysis of strawberry *bZIP* genes. The phylogenetic tree was generated using the amino acid sequences of bZIP proteins via NJ methods. All strawberry *bZIP* genes were classified into 10 groups. (**b**) Gene structure. Through the website Gene Structure Display Server, the whole genome sequence and CDS sequence of *FvbZIP* genes were formulated into an information map containing the structure of each *FvbZIP* gene. (**c**) Motif patterns of strawberry bZIP proteins. The protein sequence of all the strawberry bZIP proteins were entered into the MEME, with the highest motif number set to 20 and the other parameters set to the default.

**Figure 5 plants-09-01199-f005:**
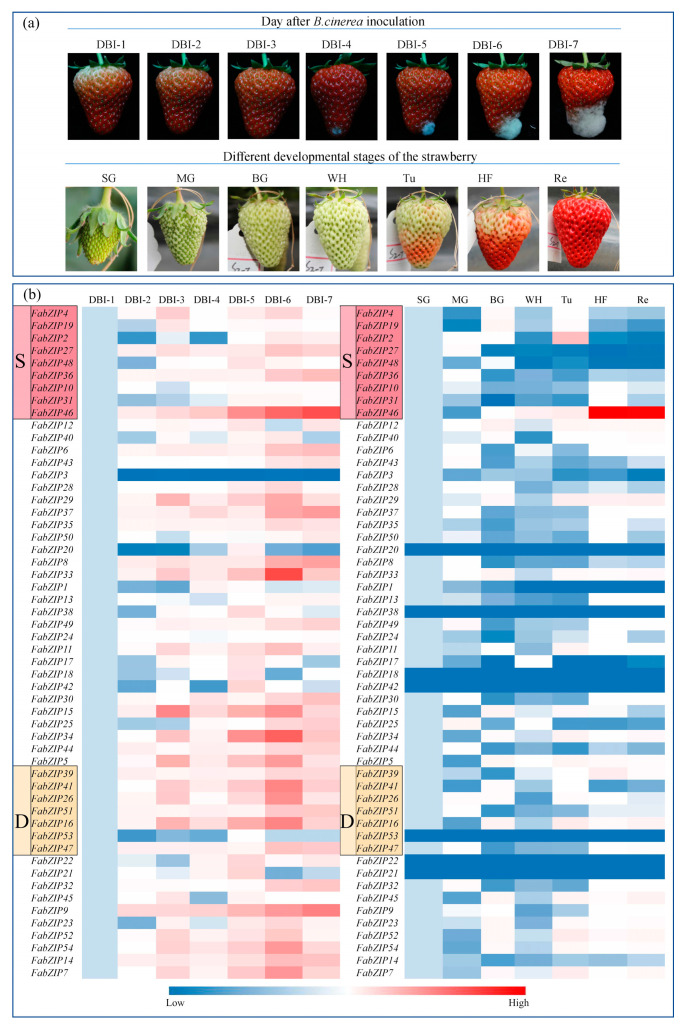
(**a**) Photographs showing the different developmental stages of strawberries. The whole process of strawberry development was divided into SG, MG, BG, Wh, Tu, HF, and Re stages. Phenotypes of red fruit after *Botrytis cinerea* inoculation. (**b**) The expression of *FabZIP* genes in different strawberry growth stages and in different stages after *Botrytis cinerea* inoculation of red fruit.

**Figure 6 plants-09-01199-f006:**
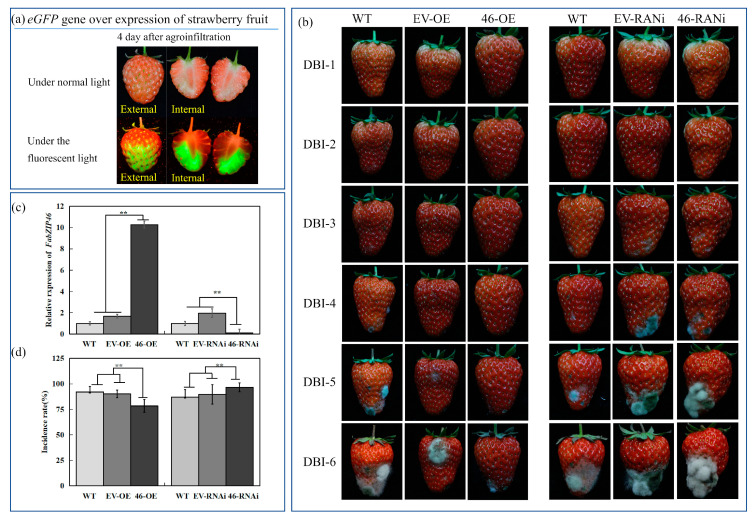
(**a**) Manipulated *FabZIP46* gene expression levels in strawberries, obtained using transient overexpression and gene silencing techniques. Green fluorescence emitted by transformed strawberries was detected at four days after injection, indicating that the *FabZIP46* gene was overexpressed or silenced in strawberries. (**b**) Phenotypes of *FabZIP46-OE* and *FabZIP46-RNAi* strawberries after *Agrobacterium tumefaciens* injection and *Botrytis cinerea* inoculation. Strawberries at the red stage of development were injected with *A. tumefaciens*, and *B. cinerea* inoculation was performed four days after *A. tumefaciens* injection. To avoid tissue damage that might affect the experimental results, approximately 1 mL (depending on the fruit size) of *A. tumefaciens* suspension was evenly injected into the fruits through the pedicel with a sterile hypodermic syringe. The injection depth was approximately half of the longitudinal diameter of the fruit to ensure that the *A. tumefaciens* suspension was fully released in the pith of the fruit. The first vertical line, second vertical line, fourth vertical line, and fifth vertical line represent the phenotypes of non-transgenic control fruits and transgenic fruits with empty vectors. The third vertical line and sixth vertical line show the phenotypes of transgenic fruits with *FabZIP46-OE* and *FabZIP46-RNAi*, respectively. (**c**) Changes in relative *FabZIP46* gene expression levels in *FabZIP46-OE* and *FabZIP46-RNAi* fruits. Values are means ± SD (*n* ≥ 3). Asterisks above the columns denote a significant difference at ** *p* < 0.05 levels according to the Student’s *t*-test. (**d**) Comparison of the incidence of *FabZIP46-OE* and *FabZIP46-RNAi* in strawberries after *Botrytis cinerea* inoculation.

**Figure 7 plants-09-01199-f007:**
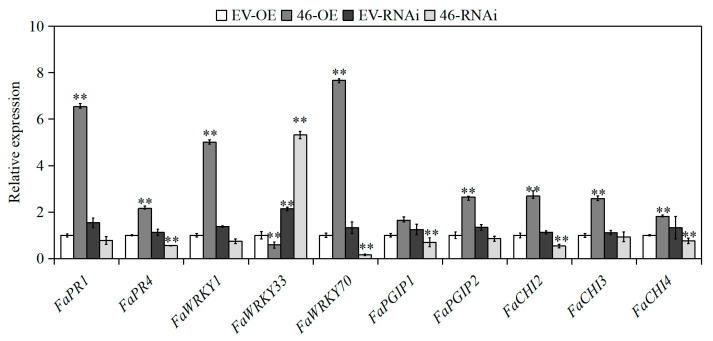
Effects of *FabZIP46-OE* and *FabZIP46-RNAi* on the transcription of resistance-related genes. *EV-OE* and *EV-RNAi* denote controls for the overexpressed or silenced fruits transformed with the corresponding empty vector. Values are means ± SD (*n* ≥ 3). Asterisks above the columns denote significant difference at ** *p* < 0.05 levels according to the Student’s *t*-test.

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
