# Peer review of "Genome-Wide Identification and Expression Analysis of the Strawberry FvbZIP Gene Family and the Role of Key Gene FabZIP46 in Fruit Resistance to Gray Mold"

_plants, 2020, doi:10.3390/plants9091199_

Round 1

Reviewer 1 Report

Comments to the Author

Bei Lu and coauthors identified 54 bZIP transcription factors from the Fragaria x ananassa genome using bioinformatic methods. Genes have been characterized in terms of structure, chromosomal location, subcellular localization, phylogeny, and involvement in the strawberry response to biotic stress – gray mold disease. A list of FabZIP genes was elucidated, the transcription of which was increased or decreased after infection with Botrytis cinerea. It has been hypothesized and then experimentally confirmed that one of these genes, FabZIP46, plays an important role in the mechanism of strawberry resistance to gray mold disease.

Strawberry is one of the most common berry crops with high nutritional qualities and, at the same time, is considered a model plant for researching fleshy fruits. Therefore, the study of the families of multifunctional transcription factors, as well as the mechanisms of stress resistance of this species, is relevant both for strawberry breeding and for studying the mechanisms of development and response to external conditions of higher plants.

The main novelty of the manuscript is in determining the role of the FabZIP46 gene in the regulation of strawberry resistance to Botrytis cinerea infection.

The manuscript may be accepted for publication in Plants after a major revision.

MAJOR COMMENTS

According to the authors (including the title of the manuscript), the research was carried out on the strawberry species Fragaria x ananassa.

Analysis of the Table S1 showed that all the genes listed in it, in accordance with their LOC numbers, belong to the Fragaria vesca, and not to Fragaria x ananassa. For instance, LOC101299496 – uncharacterized LOC101299496 [Fragaria vesca (wild strawberry)]. Moreover, the LOC101299607 that corresponds in Table S1 to FabZIP46, is “LOC101299607 basic leucine zipper 43-like [Fragaria vesca (wild strawberry)]”. Also, L479-481: “By referencing the sequence of the FabZIP46 gene published by NCBI (GenBank accession number: XM_011468884.1)” - XM_011468884.1 is basic leucine zipper 43-like [Fragaria vesca].

It is unclear whether the authors confused the names of the studied strawberry species or indeed analyzed the Fragaria x ananassa genome. It should be clarified. If the analyzed genes belong to Fragaria x ananassa, authors should deposit those genes in the NCBI or other publicly accessible scientific database and indicate the identification numbers in the manuscript.

L92: “…bZIP gene family were identified in the whole strawberry genome”. This should include the identifier of Fragaria x ananassa genome in the NCBI or in other publicly available database, as well as a reference, if available.

However, it should be noted that the Fragaria vesca bZIP gene family has already been characterized [Xiao-Long Wang et al. Genome-Wide Identification of bZIP Family Genes Involved in Drought and Heat Stresses in Strawberry (Fragaria vesca). Int J Genomics. 2017;2017:3981031. doi: 10.1155/2017/3981031.]. Wang and co-authors identified 50 FvbZIP genes, characterized their exon-intron structure, tissue-specific expression patterns, phylogenetic relationship of strawberry proteins with Arabidopsis and rice bZIP proteins, as well as protein dimerization properties and conserved motifs. Among the identified FvbZIP transcription factors, Wang et al. defined those associated with drought and heat stress responses in strawberry.

In this regard, Bei Lu with co-authors should carefully compare the results obtained with the data of Wang et al.

In another paper, Wang et al. analyzed the divergence of the bZIP Gene Family in Strawberry, Peach, and Apple [Xiao-Long Wang et al. Divergence of the bZIP Gene Family in Strawberry, Peach, and Apple Suggests Multiple Modes of Gene Evolution after Duplication. Int J Genomics . 2015;2015:536943. doi: 10.1155/2015/536943.]. This should also be discussed by Bei Lu with co-authors, as they perform phylogenetic analysis of FabZIP proteins in their study.

MINOR COMMENTS

  1. Supplementary materials such as Tables S1, S2, and S3 are not cited in the text of the manuscript. Links should be given in the chapter “Results”. For instance, L94: “…numbered from FabZIP1 to FabZIP54”. Presumably there should be a link to the Table S1 here.
  2. Please, provide a valid WEB-links or references for these programs: L460: Link http://mg2c.iask.in/mg2c_v2.0/ for MapGene2Chrom web v2 doesn’t work: ; L464: Link http://gsds.cbi.pku.edu.cn/index.php doesn’t work. 
  3. Line 14: In the abstract, the name of the species “B. cinerea” must be written in full - Botrytis cinerea
  4. L47-54: The sentence “The members of this family are mainly involved in the growth and development of plants, and they respond to biotic and abiotic stresses” in the sense of repeating the sentence “Numerous studies have indicated that members of the bZIP gene family are involved in biological processes such as growth and development, senescence, hormone regulation, energy metabolism, pathogen defense, and abiotic stress of plants [10-14]”. It would be better to remove the first sentence from the text and leave only the second one.
  5. L81: ‘revolutionary relationship’ should be replaced with ‘evolutionary relationship’.
  6. L3, 81, 137, 140, 142, 333: ‘chromosome orientation’ should be replaced with ‘chromosomal localization’.
  7. L101: ‘orientation’ should be replaced with ‘localization’
  8. The authors are very careless with the concepts of gene and protein:

L82: ‘gene domains’ should be replaced with ‘domains of the encoded proteins’.

L91-92: “…transcription factors of the bZIP gene family” - transcription factors of the bZIP family.

L98: “FabZIP31” – not italic, this is the name of the protein.

L99: “the molecular mass of the 54 FabZIP genes proteins” - the molecular mass of the 54 FabZIP proteins. “FabZIP” – not italic.

L145-151: Genes do not contain domains, but they encode proteins, which contain conserved domains. If the authors write about protein domains, they should write names in direct font. “nine FabZIP genes were found to not only contain the typical bZIP conserved domains but also other domains” – nine FabZIP proteins.

L156: “in the amino acid conserve domain of the bZIP genes of strawberry” – the bZIP proteins.

L157: “bZIP conserve domain of FabZIP genes” – FabZIP proteins.

L161: “basic domain of FabZIP32” – not italic.

L162: “rice genes OsbZIP21 and OsbZIP82” – rice proteins.

L163: “it is evident that the conserve domains of Arabidopsis and strawberry are” – “it is evident that the bZIP conserved domains of Arabidopsis and strawberry proteins are …”

L167: “FabZIP genes” – FabZIP proteins.

L169, 170, 171, 172, 173, 174: instead of gene (genes) should be protein (proteins).

L200: “strawberry bZIP transcription factors” – strawberry bZIP genes

L349: "FabZIP46" – not italic

L351-357: The authors talk about proteins – all names should not be italic.

L439-442: "The structures of the downloaded FabZIP genes were predicted using NCBI-CDD. genes that did not include the bZIP structural domain were deleted. In addition, the molecular weight and isoelectric point of the amino acid sequence of all bZIP genes were…” - Proteins!

L447: “The protein sequences of Arabidopsis bZIP genes” - The amino acid sequences of Arabidopsis bZIP proteins

L450-451: “to compare the protein sequence of the downloaded protein and that of the predicted FabZIP genes” – FabZIP proteins

L455: “protein sequence of FabZIP genes” – FabZIP protein sequences

L466-469: “Moreover, the structures of the conserved domain and other protein motifs of strawberry FabZIP genes were analyzed. The protein sequence of all the grape bZIP genes.” – FabZIP proteins, bZIP proteins.

L475: "FaACTIN" – italic

In general, the authors should carefully check the manuscript for the correct use of the terms 'protein' and 'gene'.

  1. Authors should provide references to various statements: L178-179, L282-290, L291-297, L304, 306, 320, 362, 363, and 377. Also, L228-230: “Therefore, the function of FabZIP46 may be similar to that of LrbZIP1 and CcPPI1 and may be related to the resistance of strawberry to gray mold disease”. Provide references about the functional roles of LrbZIP1 and CcPPI1.
  2. Highlight names of the species in italics:

L133-136: “At represented A. thaliana, Os represented O. sativa. The phylogenetic tree was generated using the amino acid sequences of selected bZIP genes via NJ methods. All strawberry bZIP genes, together with their A. thaliana and O. sativa homologues”

L233-234: Botrytis cinerea – italic

L265: A. tumefaciens - italic

L309: Arabidopsis thaliana - italic

  1. Figures:

Figure 3: In caption, authors should indicate which part of the proteins they used for alignment.

L193: Figure 4. Phylogenetic tree – describe how you did it (like for Fig. 1)

L199: “Figure 4” – Figure 5

L238: “Figure 5” - Figure 6

Figure 5: The S and D bZIP genes should be highlighted in the picture. Otherwise, it is very difficult to comprehend the description (Lines 210-219) of the gene expression pattern during berry development and decease progression.

L262: “FabZIP46-RANi” – FabZIP46-RNAi

L298-301: space between words.

  1. In the "discussion" chapter, there are many repetitions of the data from the "results" chapter, for instance, L381-386. Also, the data on L351-357 should be replaced into the chapter "results".
  2. The methods used are insufficiently described:

L482: “The vector was constructed using the Gateway technology”, “ TOPO vector”. What kit was used? Provide information.

L484: “LR reaction” – please, decipher.

L485: “pH7WG2D” – please, provide reference or detail description.

L489: “pFGC5941” – please, provide reference or detail description.

L494: “Agrobacterium GV3101” – Agrobacterium tumefaciens strain GV310.1 – name in italic.

L495: “freeze-thaw method” – please, provide reference or detail description.

L497: “supplementary data” – please, provide number of table or figure.

L526: “Premier” – Primer.

  1. L100: kd – kDa
  2. L109: “phylogenetic tree of strawberry, Arabidopsis, and rice was constructed” – “phylogenetic tree was constructed based on the amino acid sequences of strawberry, Arabidopsis, and rice bZIP proteins”
  3. L115-117: “The analysis revealed that 85% of the 54 FabZIP genes were more closely related to the bZIP genes of Arabidopsis, indicating that most of the FabZIP genes were more closely related to the bZIP genes of Arabidopsis than to those of rice” – The analysis showed that 85% of the 54 FabZIP proteins cluster predominantly with Arabidopsis bZIP proteins, which indicates their closer evolutionary relationship than with rice bZIP proteins.
  4. L140: ‘MapGene2Chronomose2’ – should be ‘MapGene2Chrom web v2’ or ‘MapGene2Chromosome2’
  5. L143-144: “Chromosome 2, which contained 17 of these genes, had the most 143 genes distributed on it.” – better to rewrite, for instance: “Chromosome 2 contained the largest number of genes - 17.”
  6. L177: “conserved domains” – motifs
  7. L186: “most of them had conserved loci and stages” – what does “stages” mean here?
  8. L189: “However, 11 genes were found to have no intron in FabZIP genes” - However, 11 FabZIP genes were found to have no introns
  9. L191-192: "Furthermore, the FabZIP genes have similar structures clustered relatively close together." - Moreover, the FabZIP genes with a similar structure were predominantly clustered together.
  10. L252-253: “overexpression of FabZIP46 regulates the resistance of strawberry fruits to gray mold disease” – overexpression of FabZIP46 increases the resistance of strawberry fruits
  11. L434: “physiochemical” – physicochemical
  12. L466-469: “The protein sequence of all the grape bZIP” – GRAPE? Also: what program was used? Please, provide name, reference and Web-link.
  13. L474: to synthesize cDAN – cDNA

Reviewer 2 Report

Bei Lu et. al., identified 54 FabZIP genes after bioinformatic analysis of the strawberry genome. The structure of these genes were found to be highly conserved, and the expression of FabZIP genes changed during different stages of its growth and of its infection with gray mold disease. More specifically, FabZIP46 found to play a positive role in the resistance of strawberry fruit to B. cinerea. They used modern bioinformatics tools and enriched the study using transformation and expression analysis in order to gain a better understanding of FabZIP genes role in strawberry. I suggest the manuscript to be accepted after minor revision.

Minor comments:

Grammar and language check is needed. Some expressions are repeated many times in the manuscript.

Line 238 its Figure 6

Line 199 its Figure 5

Line 304 Add relevant references

Author Response

The language and grammer has been modified and the error content has changed.

Reviewer 3 Report

The manuscript entitled 'Genome-Wide Identification, Expression Analysis and the Role of the Key Gene in the Resistance to Gray Mold Disease of bZIP Gene Family in Strawberry(Fragaria × ananassa)' is focusing their research on a very interesting and worldwide demanding fruit (Strawberry) for finding the genetic regulators to check the Gray mold disease resistance which is limiting the yield enormously. Genome wide mining of the candidate genes and their transient expression of the resistant genes are very much pilot research for the disease resistance.

Generally the manuscript is well written, explained the result scientifically and the structure is acceptable. So, I am in favour of the publication of this manuscript with minor corrections of the following 2 points:

  1. Please suggest some other title. Present title is little bit confusing for 'Gray Mold Disease of bZIP Gene Family'
  2. Authors' affiliations and their email address are mixing together. So, please organize it.

Author Response

1. The title has been changed.

2. Authors' affiliations and  email address have been organized.

Round 2

Reviewer 1 Report

The edited version of the manuscript can be accepted for publication after minor corrections. In addition, English revision is needed to improve the presentation of the data.   Comments to the Authors L117: “Arabidopsis, and rice bZIP proteins, Arabidopsis, and rice” Remove one repeat “Arabidopsis, and rice” L153: “FvbZIP35, FvbZIP37, and FvbZIP50” Not italic – here, these are the protein names! L155-156: “FvbZIP16, FvbZIP26, FvbZIP39, FvbZIP41, FvbZIP47, FvbZIP51 and FvbZIP53” Not italic – here, these are the protein names! L178-180: “the strawberry bZIP family (Figure 4). The result showed that 178 all the proteins in the strawberry bZIP family contained 1-7 motifs. FvbZIP36 exhibited only one motif; additionally, all the proteins, except FvbZIP32, exhibited” MEME motifs were found in protein sequences. Not italic – here, these are the protein names! L200: “the amino acid sequences of bZIP genes” Genes have nucleotide sequences. “The amino acid sequences of bZIP proteins” L204: “Motif patterns of strawberry bZIP genes” Motifs were found in protein sequences, not in genes! “Motif patterns of strawberry bZIP proteins” L247: “Fusarium oxysporum” In italics L248: “CcPPI1 is a nuclear protein” Not italic, protein name L338: “Arabidopsis thaliana” Italic L520-522: “inserted into the TOPO vector. TOPO PCR Cloning technology can perform benchtop cloning reactions in just five minutes with up to 95% recovery of our desired clone. TOPO vector was purchased from Thermo Fisher Scientific Inc.” Better: “inserted into the TOPO vector (TOPO PCR Cloning technology; Thermo Fisher Scientific Inc.)”

Author Response

It has been modified one by one according to the suggestions of reviewers.